# Development of a national quality framework for palliative care in a mixed generalist and specialist care model: A whole-sector approach and a modified Delphi technique

Manon S. Boddaert[1,2]*, Joep Douma[1,3], Anne-Floor Q. Dijxhoorn[1,2], René A. C. L. Héman[4], Carin C. D. van der Rijt[3,5], Saskia S. C. M. Teunissen[6], Peter C. Huijgens[1], Kris C. P. Vissers[3,7]

1 Netherlands Comprehensive Cancer Organisation (IKNL), Utrecht, the Netherlands, 2 Center of Expertise in Palliative Care, Leiden University Medical Center, Leiden, the Netherlands, 3 Palliactief, Dutch Society for Professionals in Palliative Care, Delft, the Netherlands, 4 Royal Dutch Medical Association, Utrecht, the Netherlands, 5 Department of Medical Oncology, Erasmus University Medical Center Cancer Institute, Rotterdam, the Netherlands, 6 Netherlands Association for Palliative Care (PZNL), Utrecht, the Netherlands, 7 Department of Anesthesiology, Pain and Palliative Medicine, Radboud University Medical Center, Nijmegen, the Netherlands

* m.boddaert@iknl.nl

**Data Availability Statement:** This manuscript describes the process of consensus-building and

## Abstract

In a predominantly biomedical healthcare model focused on cure, providing optimal, person-centred palliative care is challenging. The general public, patients, and healthcare professionals are often unaware of palliative care's benefits. Poor interdisciplinary teamwork and limited communication combined with a lack of early identification of patients with palliative care needs contribute to sub-optimal palliative care provision. We aimed to develop a national quality framework to improve availability and access to high-quality palliative care in a mixed generalist-specialist palliative care model. We hypothesised that a whole-sector approach and a modified Delphi technique would be suitable to reach this aim. Analogous to the international AGREE guideline criteria and employing a whole-sector approach, an expert panel comprising mandated representatives for patients and their families, various healthcare associations, and health insurers answered the main question: 'What are the elements defining high-quality palliative care in the Netherlands?'. For constructing the quality framework, a bottleneck analysis of palliative care provision and a literature review were conducted. Six core documents were used in a modified Delphi technique to build the framework with the expert panel, while stakeholder organisations were involved and informed in round-table discussions. In the entire process, preparing and building relationships took one year and surveying, convening, discussing content, consulting peers, and obtaining final consent from all stakeholders took 18 months. A quality framework, including a glossary of terms, endorsed by organisations representing patients and their families, general practitioners, elderly care physicians, medical specialists, nurses, social workers, psychologists, spiritual caregivers, and health insurers was developed and annexed with a summary for patients and families. We successfully developed a national consensus-based patient-centred quality framework for high-quality palliative care in a mixed generalist-specialist

development of a national quality framework. As such, the manuscript together with the Supporting Information Files 1-3, provide the study's minimal data set and will enable others to inform their own process of consensus-building and development of a national quality framework for palliative care. The Netherlands Quality Framework for Palliative Care has been translated into English and is freely available at www.palliaweb.nl/publicaties/netherlands-quality-framework-for-palliative-care. The qualitative data (feedback from expert panel and consultation of peers) on which the content of the Netherlands Quality Framework for Palliative Care is based, are in Dutch and available from the Netherlands Comprehensive Cancer Organisation (IKNL) upon request (contact via info@iknl.nl).

**Funding:** The author(s) received no specific funding for this work.

**Competing interests:** I have read the journal's policy and the authors of this manuscript have the following competing interests: CCDR declares receiving fees from Kyowa Kirin for consulting activities outside the submitted work and paid to her department. The other authors have declared that no competing interests exist. This does not alter our adherence to PLOS ONE policies on sharing data and materials.

palliative care model. A whole-sector approach and a modified Delphi technique are feasible structures to achieve this aim. The process we reported may guide other countries in their initiatives to enhance palliative care.

## Introduction

Palliative care aims to improve the quality of life of patients, and their families, who are facing problems associated with life-threatening illnesses, through the prevention and relief of suffering by means of early identification, impeccable assessment, and the treatment of pain and other physical, psychosocial, and spiritual problems [1]. Palliative care is frequently delivered by generalists in palliative care, for issues such as advance care planning in a primary care setting or symptom management in secondary care [2, 3]. In addition, multidisciplinary specialist palliative care teams deliver care for more complex needs in inpatient, outpatient, or community-based service models [4, 5].

However, within a predominantly biomedical healthcare model focused on cure, it is challenging to provide optimal, person-centred palliative care grounded in comfort and dignity [6–9]. The general public, patients, and healthcare professionals are frequently unaware of the benefits of palliative care and how and when to access it [10, 11]. Moreover, patients in a palliative care trajectory face challenges brought about by the disease as well as by complicated and fragmented healthcare systems, which require coordination between healthcare professionals, various healthcare settings, and diagnostic and treatment interventions [12–14]. Additionally, most healthcare professionals lack sufficient training and skills in symptom management, communication, and care coordination [15, 16]. Poor interdisciplinary teamwork and limited communication combined with a lack of early identification of patients with palliative care needs contribute to the provision of sub-optimal palliative care [17–19]. Therefore, patients in a palliative care trajectory continue to receive inappropriate treatments at the end of their lives, often leading to poor quality and high-cost care [6, 20, 21]. This is despite evidence that the early integration of generalist and specialist palliative care improves symptoms, the quality of life, and quality of care for these patients [22–30], and notwithstanding professional organisations' recommendations for earlier and routine co-management by palliative care specialists [31–33].

In 2014, the World Health Organization (WHO) called for standardised availability, equitable access, and high-quality palliative care as a human right and the strengthening of generalist and specialist palliative care as a component of integrated care throughout the patient's life [34]. In high-income countries, approximately 75% of people approaching the end of their lives could benefit from palliative care and even more are expected to need palliative care in the future [35–37]. To anticipate this foreseen increase, and an unforeseen tsunami of suffering as witnessed during the COVID-19 pandemic, healthcare systems need to focus on the integration of palliative care across all levels of health and social care disciplines, while preparing and training all healthcare professionals to deliver generalist palliative care [3, 4, 38–41].

In the Netherlands, national palliative care programmes have been part of the government's health policy since 2007, and a white paper and a standard for palliative care have since been developed [42, 43]. However, concerns regarding life-prolonging treatments prevailing over comfort-oriented care near the end of life remain [44]. With the intent to improve availability and access to high-quality palliative care for all people with life-threatening illnesses, we developed a national consensus-based quality framework for the optimal organisation and delivery

of patient-centred palliative care in a mixed generalist and specialist palliative care model [45]. We hypothesised that a whole-sector approach and a modified Delphi technique could be beneficial for the broad recognition and integration of palliative care [19, 46]. The process of development and consensus-building and its key elements are presented here.

## Methods

For the development of this national quality framework for palliative care, we adhered to the Guideline for Guidelines [47], a complementary tool to the revised international criteria for Appraisal for Guidelines of Research and Evaluation (AGREE II) [48]. Considering the broad scope and the multidisciplinary nature of palliative care as well as an extensive amount of stakeholders, we employed a whole-sector approach [19] and consulted an expert panel in a modified Delphi technique to answer the research question [49–51]. We structured the development into three phases: Preparation, development, and finalisation (Fig 1).

### Preparation—Building consensus and an organisational structure

**a. Research team of senior peers in palliative care.** In late 2014, a research team of senior peers in palliative care (senior physicians in palliative care and medical oncology) representing the Dutch Society of Professionals in Palliative Care (Palliactief) and the Netherlands Comprehensive Cancer Organisation (IKNL), initiated the development of the Netherlands Quality Framework for Palliative Care (NQFPC). The role of the research team consisted of planning and managing the overall project, consulting and informing the stakeholders and processing the results from the Delphi rounds to inform each next step in the process.

**b. Stakeholder involvement.** The research team consulted 26 stakeholder associations and organisations. In line with a whole-sector approach, they represented various medical, nursing, and allied health professional disciplines, patients, informal caregivers, volunteers,

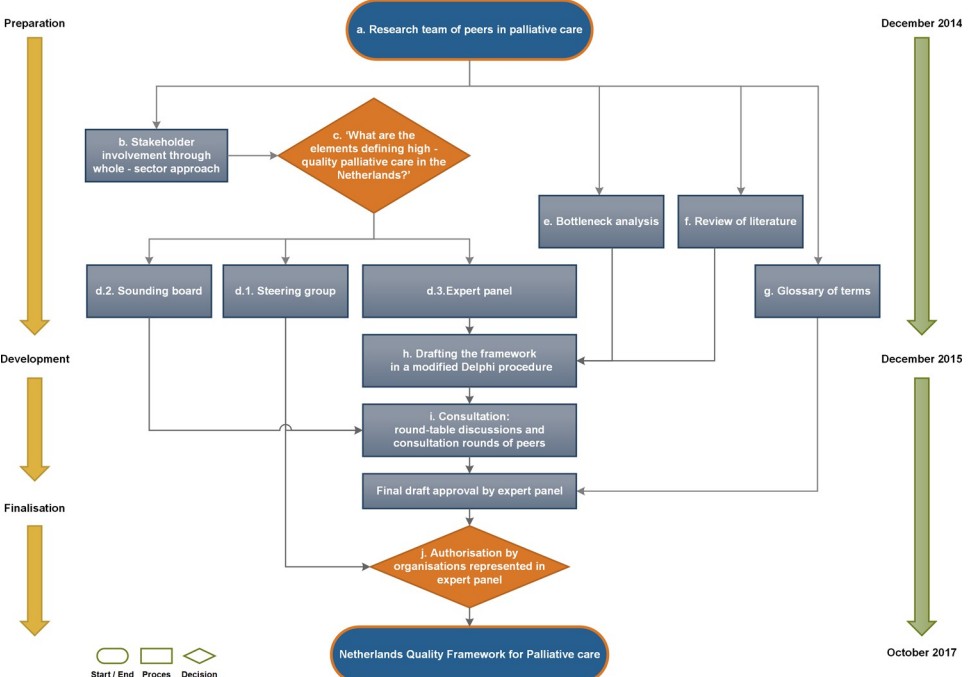

**Fig 1. Structure and process to develop the quality framework.** Letters a-j refer to subheadings in text.

health insurers, healthcare umbrella organisations, hospice care organisations, and policy-makers (S1 Appendix). In personal interviews, the research team explored their views on the need for and their willingness to contribute to the development of a national consensus-based quality framework in order to obtain whole-sector support.

**c. Main question.**   Broad consent was acquired, and stakeholders agreed to answer the main question: 'What are the elements defining high-quality palliative care in the Netherlands?'

**d. Structure for the development of the NQFPC.**   To answer this main question, the consulted stakeholder organisations were invited to participate in: 1) a steering group (organisations representing the patient population, the targeted users of the quality framework, and health insurers) or 2) a sounding board (organisations representing patients with specific diagnoses, hospice care organisations, policymakers, research institutes, among others), to support 3) an expert panel for the duration of the development process.

This multidisciplinary expert panel represented patients, healthcare providers, and health insurers, and was formed to draft the content of the NQFPC. Ten members of the expert panel represented the targeted users of the quality framework and originated from all regions of the Netherlands–various types of hospitals, relevant care settings, and disciplines. They had either generalist or specialist expertise in palliative care, in addition to expertise in anaesthesiology/ pain medicine, internal medicine, medical oncology, geriatrics, primary family healthcare, elderly healthcare, nursing, psychology, social care, and spiritual care and were mandated by their national organisations to provide their inputs. Two members of the expert panel were representatives of patients. To ensure that the content of the NQFPC would primarily be care- and quality-driven and that costs-related interests would be secondary, the representative of all Dutch health insurers did not participate in expert panel meetings but provided input in subsequent separate meetings with the research team.

## Development—The process of drafting the content

**e. Bottleneck analysis.**   For the NQFPC to improve the accessibility and availability of high-quality palliative care, perceived bottlenecks or barriers in palliative care practice needed to be identified in order to be addressed in the quality framework. Therefore, a search of Dutch palliative care literature published between 2005 and 2015 was performed [52]. Furthermore, a national survey assessing the organisation and quality of specialist palliative care in hospitals was conducted. The results of both were discussed in an invitational conference with representatives from medical and nursing organisations in primary care, initiated by the Royal Dutch Medical Association. Attendees were asked to (1) indicate whether they agreed with the identified barriers; (2) suggest potential solutions for daily practice; and (3) indicate additional problems [52].

**f. Review of literature.**   The aim of the literature review was to identify international quality reports, guidelines, frameworks, and standards for palliative care that could serve as core documents for the development of the Dutch quality framework. Search terms consisted of 'quality standards AND palliative care AND hospice care'. As PubMed and Google Scholar hardly provide results for published standards of care, we performed our search in the Google database [53]. Titles were screened as the first step in the assessments of potentially relevant results. Subsequently, reports, guidelines, frameworks, and standards describing criteria for palliative care were included and manuscripts, books, and websites were excluded. The remaining documents were evaluated by their cover page and included according to the date of publication (between 2005 and 2015), and when publications were in English, documents originated from high-income countries, and the content had a national scope. Subsequently,

documents were analysed and those with abstracts or summaries referring to all patients with palliative care needs (i.e., they were not limited to specific diseases or patient-groups) and with cross-references to scientific literature were included. Finally, the content of the remaining documents was reviewed to ascertain whether it comprised the entire scope of palliative care and was applicable to the Dutch healthcare setting. A similar strategy was employed for a review of national literature.

**g. Glossary of terms.** During stakeholder consultations, it became apparent that a mutual understanding of terminology was needed. Therefore, a glossary of terms was added to the quality framework. While constructing the framework, terms for which clarification was deemed important were identified. Definitions of these terms were searched in national and international literature. When no definition seemed available, or if it did not fit the context of the quality framework, experts in the discipline concerned were consulted to formulate a consensus-based definition or description of the term.

**h. Drafting the quality framework in a modified Delphi procedure.** To answer the research question, we used a modified Delphi technique among the members of the expert panel. This technique is based on gathering the experts together and discussing the issues from a Delphi survey round in a structured way to reach consensus among all participants simultaneously. Thus, the modified Delphi technique can achieve consensus in a more time- and cost-effective manner [50]. We alternated two written Delphi survey rounds with face-to-face meetings of the expert panel [49–51].

Analogous to the predominant structure of the core documents, the expert panel constructed the quality framework using domains, standards, and criteria. Each domain described a dimension of palliative care and consisted of one or several standards indicating best practice, supported by several criteria on how to achieve these standards. For each domain, expert panel members were invited to suggest additional national literature that could aid in tailoring the quality framework to the Dutch healthcare system.

*Data extraction*. In the initial face-to-face meeting with the expert panel, the selected core documents from the literature review were presented and the format for informational input in the Delphi survey rounds was piloted for clarity and feasibility. For each domain, the research team selected all relevant standards from the core documents, aligned all supporting criteria, and presented them to the expert panel in a first Delphi round (S2 Appendix). Each panel member was asked to indicate which of the standards and criteria should be included in the quality framework. Using standardisation percentages, each standard and criterion was graded for admission (> 66% agreed), discussion (50%-66% agreed), or dismissal (<50% agreed).

*Data synthesis*. From the results of the first Delphi round, the research team constructed each domain with the standards and criteria that were accepted or needed to be discussed. Subsequently, the selected standards and criteria were extensively discussed in intermediate face-to-face meetings with all expert panel members and either accepted, discarded, revised, or adapted to the Dutch context. The representatives of patients in the expert panel had the decisive vote whenever a consensus could not be reached.

As a next step, the research team processed the results from the face-to-face meetings and issued a second Delphi round with the expert panel for iterative feedback.

## Finalisation

**i. Consultation.** The research team organised two round-table discussions to inform and involve the steering group and sounding board in the drafting process. Both round-table discussions were followed by written consultation rounds among peers to gather feedback on

draft recommendations and assess applicability in clinical practice. These consultation rounds were issued at the same time as the second Delphi round with the expert panel.

**j. Authorisation.** After processing feedback from the second Delphi round and the consultation round and obtaining the approval of the final draft of the NQFPC, including its glossary of terms, in a last meeting with the panel members, it was submitted to the associations and organisations represented in the expert panel for final review and authorisation or approval.

## Ethical approval

Within the scope of the Dutch Medical Research Involving Human Subjects Act (WMO) and according to the Central Committee on Research involving Human Subjects (CCMO) this type of study is exempt from approval of an ethics committee. For more information on local legislation please see https://english.ccmo.nl/investigators/legal-framework-for-medical-scientific-research/your-research-is-it-subject-to-the-wmo-or-not and https://english.ccmo.nl/investigators/additional-requirements-for-certain-types-of-research/non-wmo-research.

## Results

It took the research team one year of preparation and building of stakeholder relationships to acquire broad consent for the development of the NQFPC from the whole sector involved in palliative care while simultaneously performing a bottleneck analysis and reviewing literature for core documents (Fig 1). In addition, slightly over 18 months were dedicated to the surveying of and convening with the expert panel, writing the NQFPC drafts, discussing content, consulting peers, and obtaining final consent from everyone involved. For clarity, our results focus on the process of defining and finalising the content of the NQFPC.

### Development—The process of defining the content

**Bottleneck analysis.** The main barriers identified as elements for improvement in the organisation and delivery of palliative care were: 1) information and communication about prognosis, treatment options, and the end of life, 2) coordination and the continuity of care, 3) expertise, education, and training of healthcare professionals, and 4) rules, regulations, and reimbursement (S3 Appendix). The first three barriers were identified by both patients and healthcare professionals and were addressed throughout the construction of the NQFPC. The last barrier was predominantly reported by healthcare professionals and was addressed separately in a supplementary guide [54]. The full results from the hospital survey and the bottleneck analysis are presented elsewhere [16, 52, 55].

**Review of literature.** The predefined search terms yielded 680,000 results in the Google database. The first 70 titles were considered eligible for initial review, as the relevance of titles increasingly diminished until the saturation of relevant results occurred. Subsequently, 40 manuscripts, books, and websites were excluded and 30 reports/guidelines/frameworks/standards describing criteria for palliative care were included (Fig 2). Assessment of the remaining documents according to the predefined procedure resulted in four international documents with content applicable to the Dutch healthcare setting and which comprised the entire scope of palliative care. A similar strategy was used to review Dutch literature. The initial search yielded 28,000 results. After applying the predefined steps to 70 initial titles, two Dutch documents remained. Consequently, four international and two national documents formed the core documents for the development of the NQFPC [43, 56–60].

**Glossary of terms—Definition of palliative care.** A glossary of terms was annexed to the NQFPC. Definitions of 82 terms that were identified as requiring clarification were searched

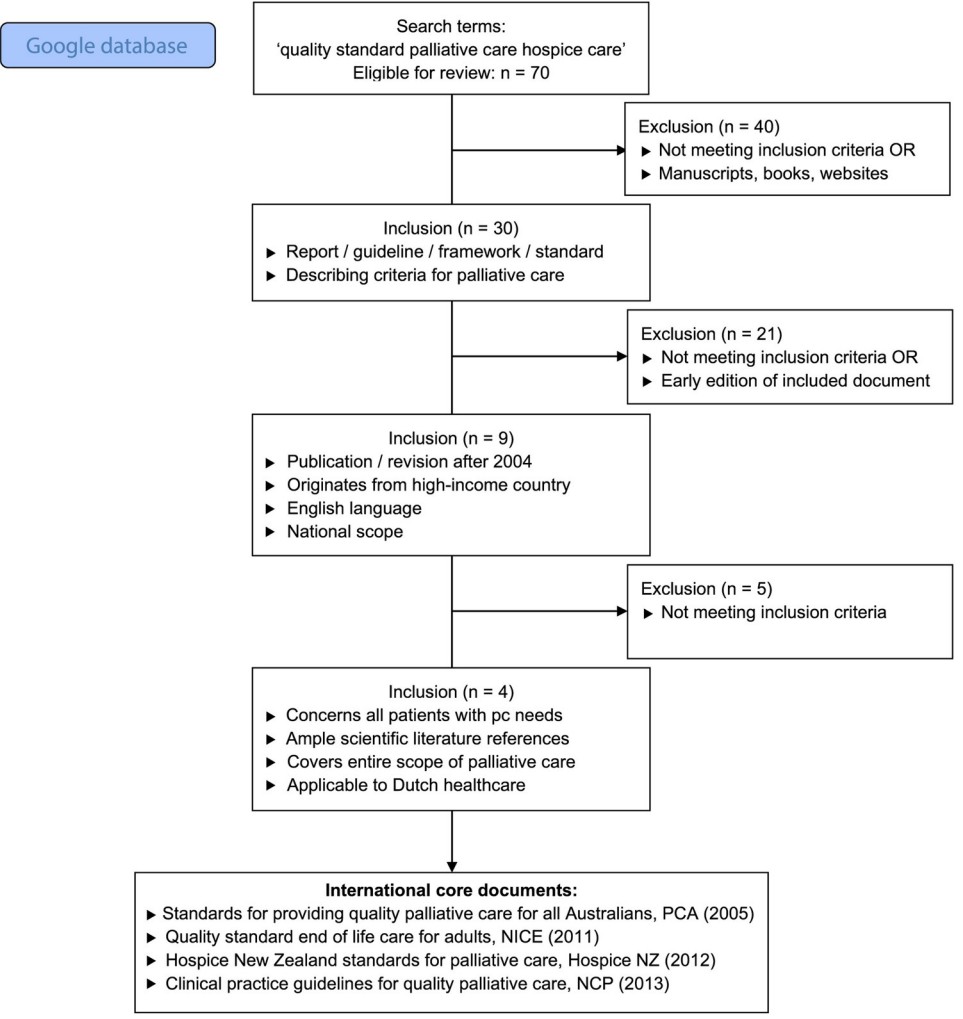

**Fig 2. Flow diagram for the review of international literature.**

and found mostly in national and international literature. Since consensus on the definition of palliative care was an important starting point to answering the main question, extensive attention was paid to it at the start of the process.

Both the expert panel and their peers (in consultation rounds) extensively discussed the 2002 WHO definition of palliative care [1]. First, this definition was compared to numerous other definitions from international literature, particularly the one used in the National Consensus Project for Quality Palliative Care (2013) [59, 61–63]. Unlike the WHO definition, this definition explicitly cited aspects of palliative care that were considered important in Dutch culture, such as interdisciplinary collaboration, dignity, autonomy, access to information, and the opportunity to make choices [64]. Conversely, the WHO definition clearly lays more emphasis on the importance of early identification. Second, the expert panel re-examined the concept of a life-threatening condition. The prevailing view was that this description did not sufficiently reflect the current diversity of the targeted patient groups within palliative care and particularly failed to include the concept of frailty. Finally, we considered it relevant to the definition that palliative care in the Netherlands is mostly delivered by generalists or non-specialists in palliative care, who receive support from specialists in palliative care when required.

Palliative care is care improving the quality of life of patients and their families, who are facing a life-threatening condition or frailty, through prevention and relief of suffering by means of early identification and careful assessment, and treatment of problems of a physical, psychological, social and spiritual nature. Over the course of the illness or frailty, palliative care aims to preserve autonomy, access to information and the opportunity to make choices.

Palliative care has the following characteristics:

- care can be given concurrently with disease-oriented treatment;
- generalist healthcare professionals and, when required, specialist healthcare professionals and volunteers, work together as an interdisciplinary team in close collaboration with patients and their families, and tailor treatment to the stated values, wishes and needs of the patient;
- to ensure continuity, care is coordinated by a central healthcare professional;
- the wishes of patients and their families concerning dignity are acknowledged and supported throughout the illness or frailty, during the process of dying and after death.

**Fig 3. Palliative care as defined in the quality framework (adapted from WHO, 2002).**

The expert panel ultimately agreed to add these important points for palliative care, in the Netherlands, to the WHO definition (Fig 3).

**Drafting the framework in a modified Delphi procedure.** The initial expert panel meeting focused primarily on evaluating and discussing the WHO definition of palliative care (Fig 3). Additionally, based on the predominant structure of the core documents, they agreed that the NQFPC should comprise nine domains consisting of standards and criteria, which together cover the entire spectrum of care for patients with a life-threatening illness or frailty and their families. These specific domains were to be preceded by a primary domain that addresses the 'core values and principles' of palliative care (Fig 4). Moreover, the expert panel members unanimously agreed to base the NQFPC on the values, wishes, and needs of patients and their families and address the barriers that had resulted from the bottleneck analysis. Furthermore, they suggested prioritising the standards and criteria specifically aimed at resolving these barriers as key elements for integration.

**Data extraction and synthesis.** The research team extracted 9 domains, 93 standards, and 626 criteria from the core documents and aligned relevant standards and criteria per domain for evaluation by each expert panel member. Based on the results from the first Delphi round, the research team constructed each domain with the standards and criteria that were either accepted by expert panel members (> 66% agreed) or needed to be discussed (50%–66% agreed). The constructed domains were evaluated and discussed, and the selected standards and criteria were either accepted, discarded, revised, or adapted to the Dutch situation across five expert panel meetings. The patients' representatives attended all expert panel meetings, actively participated in the discussions, and optimised the formulation of the patient's perspective in the draft texts. The first NQFPC draft consisted of 10 domains, 22 standards, and 137 criteria.

## Finalisation

**Consultation and authorisation.** In concurrence with the second Delphi round, two round-table discussions with the steering group and the sounding board followed by written

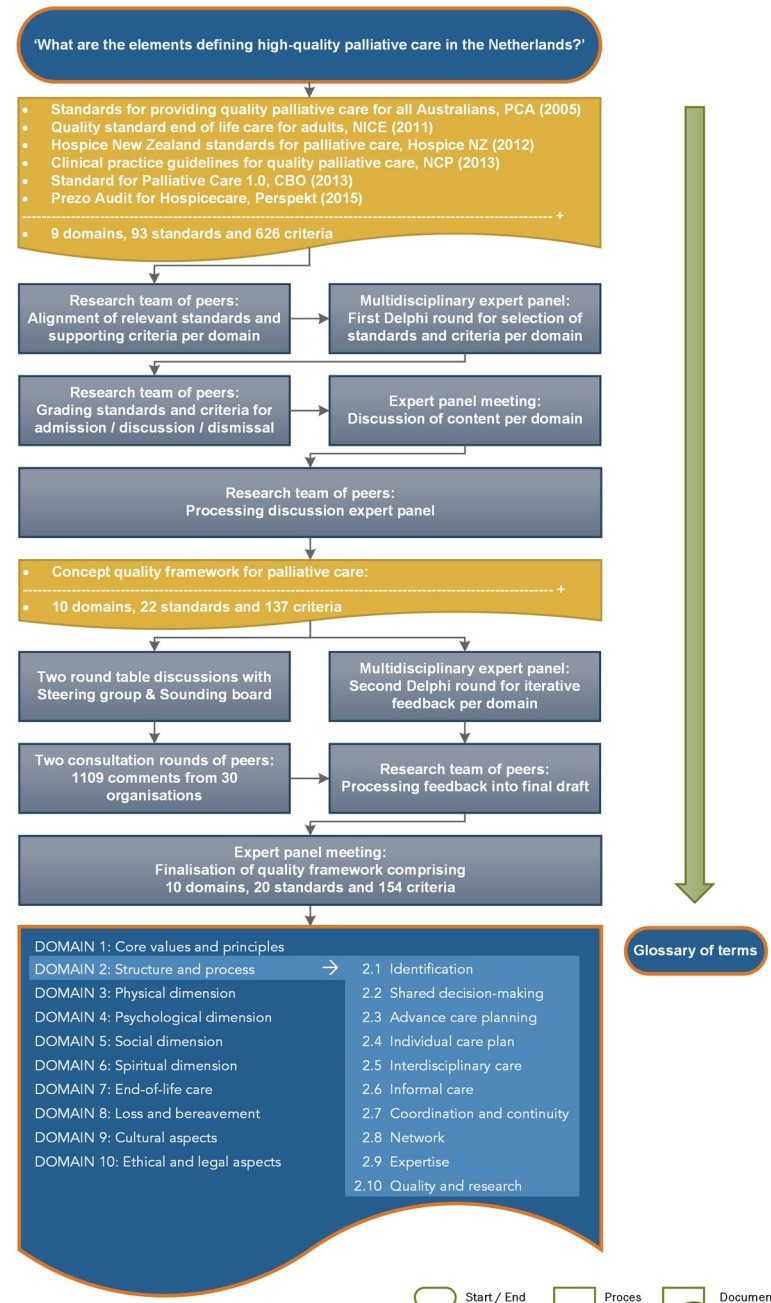

**Fig 4. Developing the content of the quality framework including a glossary of terms.**

consultation rounds with peers (Fig 4) yielded 1109 comments from 30 organisations, which were processed by the research team. In the last expert panel meeting, the final draft consisting of 10 domains, 20 standards, and 154 criteria was approved by all expert panel members. It was subsequently submitted for authorisation or approval to the associations and organisations represented in the expert panel. The NQFPC was endorsed by eight branches and umbrella organisations and was published online in October 2017. The complete framework has been translated into English and is freely available at www.palliaweb.nl/publicaties/netherlands-quality-framework-for-palliative-care.

**Table 1. Key elements in the quality framework that address barriers from the bottleneck analysis.**

| Barrier | Key element* | Originating Domain in NQFPC |
|---|---|---|
| **Information and communication (prognosis, treatment, end of life)** | Effective Communication | 1. Core values & Principles |
| | 2.1 Identification | 2. Structure & Process |
| | 2.2 Shared Decision Making | |
| | 2.3 Advance Care Planning | |
| **Coordination and continuity of care** | 2.4 Individual Care Plan | 2. Structure & Process |
| | 2.7 Coordination & Continuity | |
| **Expertise, education and training of healthcare professionals** | Work–Life Balance | 1. Core values & Principles |
| | 2.9 Expertise | 2. Structure & Process |

*A key element consists of a specific standard and their affiliated criteria as described in the Netherlands Quality Framework for Palliative Care (numbers in this table correspond to numbers in Fig 4).

With the help of the patients' representatives, a Netherlands Patients Federation editor, the Royal Dutch Medical Association, and the Netherlands Association for Palliative Care, the content of the quality framework was 'translated' into an e-book for patients [65]. This information is available at palliatievezorg.patientenfederatie.nl, a national website that provides information about palliative care for patients. The key elements (Table 1) were recommended as priorities for integration. In adherence to the Guideline for Guidelines [47], the NQFPC is intended to be updated within five years.

## Discussion

A national quality framework for palliative care seemed an essential step to optimise palliative care for the increasing numbers of patients in a highly fragmented health care system that focuses on cure rather than care. We aimed at a whole-sector approach to obtain broad consent and recognition for high-quality, patient-centred palliative care that could be integrated across health care settings. By inviting patients, healthcare professionals from various medical, nursing, and allied disciplines, health insurers, and policymakers to participate and by building this framework with a modified Delphi technique along the international AGREE II criteria (S4 Appendix), we combined the aspirations, information, resources, knowledge, and skills of all stakeholders with a scientifically sound structure and reached consensus for the NQFPC that we believe none of the parties concerned could have achieved independently [48, 49, 66, 67].

The NQFPC aims to improve the availability of equitable access to high-quality palliative care for all people with life-threatening illnesses or frailty and their families. As identified by both patients and healthcare professionals in our bottleneck analysis (S3 Appendix), barriers to achieving the above-stated standards are frequently recurring themes in international literature [12, 68–74]. In adherence to patients' values, wishes, and needs [70–72, 75], we selected the standards in the NQFPC that address these barriers as key elements for integration (Table 1) and recommended their prioritised integration in clinical practice: 1) early identification of patients in a palliative care trajectory [18], 2) shared decision making and advance care planning [69, 70, 72], 3) coordination and continuity of care, including an (electronically available) individual care plan [12, 68, 74], 4) education and training of healthcare professionals [15], that includes effective communication [69], and learning how to cope with the emotional impact of providing palliative care in order to maintain a healthy work-life balance [76].

From their initial involvement with the development of the NQFPC, the national government has been supportive of integrating the recommended key elements into regular

healthcare. A national public health campaign was initiated to raise awareness for palliative care and its benefits for seriously ill people. Furthermore, the Netherlands Organisation for Health Research and Development (ZonMw) began assessing requests for the funding of palliative care projects based on their relevance to the integration of key elements. It has since funded the whole-sector development of an educational framework for palliative care across all levels of healthcare education to prepare and train all future healthcare professionals in generalist palliative care [77]. In addition, it supported the development of a national information database to evaluate the quality of end-of-life care and establish best practice performance standards in the near future. Currently, the Royal Dutch Medical Association has adopted guardianship of the NQFPC, and various medical associations plan a step-by-step integration. Moreover, because of the NQFPC development process, we were able to apply a similar approach to developing a multidisciplinary guideline for advance care planning across healthcare settings during the COVID-19 pandemic [78, 79]. Considering these developments, we conclude that using the structure of a whole-sector approach and a modified Delphi technique not only brought broad consensus for the NQFPC content but also contributed to the awareness, recognition, and integration of palliative care in public health.

Comparing the final content of the NQFPC to the four international core documents we used in the Delphi procedure confirms that the barriers established in our bottleneck analysis are similarly perceived internationally [56–59]. All four documents address the importance of early identification, shared-decision making and care planning, coordination of care, and the training of healthcare professionals. Similar to the NQFPC, all three Anglo-Saxon documents were aimed at both generalists and specialists in palliative care and volunteers, whereas the framework from the United States (US) mainly addressed specialists in palliative care. We incorporated the structure and part of the definition of palliative care from the US framework, built a glossary of terms similar to the New Zealand document, and shared the comprehensiveness and the whole-sector approach with the Australian framework. In addition, we deemed it appropriate to address advance care planning as an individual standard in the NQFPC, as it is Dutch government policy to facilitate 'dying in the patient's preferred location' for all citizens [64].

## Strengths

In addition to the strength of a whole-sector approach, we believe that building a research team with dedicated senior physicians in palliative medicine and medical oncology was critical to facilitate the entire process. This assured the analysis of the core documents to be grounded in appropriate clinical practice and expert panel meetings to be focused on content and the organisation of palliative care. Moreover, the ease of peer consultations when the need occurred enabled us to retain ongoing support from the required medical associations and healthcare organisations.

## Conclusions

A whole-sector approach using the international AGREE II criteria and a modified Delphi technique to define the content is a feasible, effective, and efficient way to develop a national consensus-based patient-centred quality framework for high-quality palliative care. Considering the call to action from the WHO, the process described in this study contains potentially transferable information on how to develop such a framework by taking an inclusive approach and involving stakeholders from civil society rather than regarding palliative care merely as a medical discipline. As such, it may guide other countries' initiatives to improve the

accessibility and availability of palliative care and can contribute to the recognition and integration of such care in public health.

However, some limitations of the project need mentioning. Although we described the 'what' elements and defined the optimal organisation and delivery of high-quality palliative care, we did not address 'how' these elements can be realised in clinical practice or what conditions are required to build a sustainable generalist and specialist palliative care service model [3]. Similar to the Anglo-Saxon quality frameworks, the next edition of the NQFPC may be extended with clear criteria and training requirements for specialists in palliative care [56–58]. This will aid the integration and availability of palliative care by enabling workforce planning and allowing for clear and efficient interdisciplinary cooperation and reimbursement structures [2, 80]. Second, while stakeholders concerned with palliative care for people with special needs did participate as a sounding board, we limited the scope of this primary edition of the NQFPC and did not specifically address these populations. In view of equitable access to palliative care, the scope of the next edition of the NQFPC needs to include them.

While a substantial body of evidence exists to support clinical practice for quality palliative care, the quality of evidence is still limited. Hence, whether the integration of the key elements of the NQFPC in clinical practice will effectively diminish the perceived barriers for patients in a palliative care trajectory and their families is a subject that needs to be addressed through future research.

## Supporting information

**S1 Appendix. Stakeholder associations and organisations involved in development of the Netherlands quality framework for palliative care.**
(PDF)

**S2 Appendix. Delphi survey for end-of-life care domain in Netherlands quality framework for palliative care.**
(PDF)

**S3 Appendix. Results from the bottleneck analysis of palliative care provision.**
(PDF)

**S4 Appendix. AGREE reporting checklist NQFPC.**
(PDF)

## Acknowledgments

The authors wish to thank all stakeholders involved for their contributions to the development of the NQFPC; we are particularly in gratitude to the expert panel members for their time, enthusiasm, and expertise, to Birgit Frhleke (Netherlands Comprehensive Cancer Organisation, IKNL) and Wim Jansen (Palliactief, Dutch Society for Professionals in Palliative Care) for accommodating this project within their organisations, to Maureen Bijkerk for her work in researching the glossary of terms and to Elske van der Pol for managing this project and its team of peers. We would like to thank Editage (www.editage.com) for English language editing.

## Author Contributions

**Conceptualization:** Manon S. Boddaert, Joep Douma, Carin C. D. van der Rijt, Saskia S. C. M. Teunissen, Peter C. Huijgens, Kris C. P. Vissers.

**Data curation:** Manon S. Boddaert, Joep Douma, Anne-Floor Q. Dijxhoorn.

**Formal analysis:** Manon S. Boddaert, Joep Douma, Anne-Floor Q. Dijxhoorn.

**Investigation:** Manon S. Boddaert, Joep Douma.

**Methodology:** Manon S. Boddaert, Joep Douma, Anne-Floor Q. Dijxhoorn.

**Project administration:** Manon S. Boddaert, Anne-Floor Q. Dijxhoorn.

**Resources:** René A. C. L. Héman, Carin C. D. van der Rijt, Saskia S. C. M. Teunissen, Peter C. Huijgens, Kris C. P. Vissers.

**Supervision:** René A. C. L. Héman, Peter C. Huijgens, Kris C. P. Vissers.

**Validation:** Manon S. Boddaert, Joep Douma, Anne-Floor Q. Dijxhoorn, Carin C. D. van der Rijt, Saskia S. C. M. Teunissen.

**Visualization:** Manon S. Boddaert.

**Writing – original draft:** Manon S. Boddaert, Joep Douma, Anne-Floor Q. Dijxhoorn.

**Writing – review & editing:** Manon S. Boddaert, Joep Douma, Anne-Floor Q. Dijxhoorn, René A. C. L. Héman, Carin C. D. van der Rijt, Saskia S. C. M. Teunissen, Peter C. Huijgens, Kris C. P. Vissers.

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
