## [Decision Letter · Decision Letter 0]

16 Dec 2021

PONE-D-21-32051Development of a national quality framework for palliative care in a mixed generalist and specialist care model: A whole-sector approach and a Delphi procedurePLOS ONE

Dear Dr. Boddaert,

Thank you for submitting your manuscript to PLOS ONE. I am pleased to report that we were able to find a second reviewer faster than expected, so now have the requisite number of reviews available. The reviewers have made some useful suggestions for improving your paper, and we invite you to submit a revised version of the manuscript that addresses the points raised.

Would a deadline of Jan 30 2022 11:59PM be manageable with your other commitments? As I mentioned in my earlier email, feel free to request more time if needed - just contact the journal office at plosone@plos.org. Please include the following items when submitting your revised manuscript:A rebuttal letter that responds to each point raised by the academic editor and reviewer(s). You should upload this letter as a separate file labeled 'Response to Reviewers'.A marked-up copy of your manuscript that highlights changes made to the original version. You should upload this as a separate file labeled 'Revised Manuscript with Track Changes'.An unmarked version of your revised paper without tracked changes. You should upload this as a separate file labeled 'Manuscript'.

I should add that I will be on leave from 17th December to 3rd January inclusive, so please direct any questions to the PLOS ONE office in the meantime.

We look forward to receiving your revised manuscript in due course.

Kind regards,

Tim Luckett

Academic Editor

PLOS ONE

“I have read the journal's policy and the authors of this manuscript have the following competing interests: KCDR declares receiving fees from Kyowa Kirin for consulting activities outside the submitted work and paid to her department. The other authors have declared that no competing interests exist.”

Reviewers' comments:

Reviewer's Responses to Questions

**Comments to the Author**

1. Is the manuscript technically sound, and do the data support the conclusions?

Reviewer #1: Yes

Reviewer #2: Partly

2. Has the statistical analysis been performed appropriately and rigorously? 

Reviewer #1: N/A

Reviewer #2: I Don't Know

3. Have the authors made all data underlying the findings in their manuscript fully available?

Reviewer #1: Yes

Reviewer #2: No

4. Is the manuscript presented in an intelligible fashion and written in standard English?

Reviewer #1: Yes

Reviewer #2: Yes

5. Review Comments to the Author

Reviewer #1: Thank you for the opportunity to review this interesting paper. It is exceptionally well written and provides a clear and coherent account of the complex process of developing a national quality framework for palliative care in the Netherlands. There is important and potentially transferable information on how to develop such a framework taking an inclusive approach. It is helpful to see that their vision included seeing the framework as including civil society rather than regarding palliative care merely as a medical discipline.

There are a few minor points that would benefit from attention:

Page 6 Line 147-148 and Pages 9-10 The authors refer to a search of the palliative care literature. It is not clear if this literature review is already published?

Page 11-12 There is little information provided about the Delphi procedure. It is not clear how many 'rounds' were conducted.

Page 13 Table 1 and Line 332 The term 'personal balance' is referred to in the table and in the text. I wonder if this term needs to be defined?

Page 15 Lines 362-363 The authors argue that a strength of their work was building a project team of 'dedicated senior physicians in palliative medicine and medical oncology was critical to facilitate the entire process'. While I can understand that gaining the support of senior clinical leaders is necessary, their statement appears to exclude senior nurses, social workers and allied health professionals. Normally palliative care is delivered by a multidisciplinary team. Perhaps the authors can clarify if health professionals beyond medicine were excluded.

Page 15 Lines 373-374 I realise there is a word limit, but I think the authors undersell the potential transferability of their experience in developing this quality framework.

Reviewer #2: please see attachment -

6. PLOS authors have the option to publish the peer review history of their article (what does this mean?). If published, this will include your full peer review and any attached files.

Reviewer #1: No

Reviewer #2: No

---

## [Author Response · Author response to Decision Letter 0]

9 Feb 2022

My response to reviewers is added in a separate file. As suggested by the Academic editor, I updated both the Competing interests statement and the Data availability statement in the cover letter added in the uploaded files.

---

## [Decision Letter · Decision Letter 1]

8 Mar 2022

Development of a national quality framework for palliative care in a mixed generalist and specialist care model: A whole-sector approach and a modified Delphi technique

PONE-D-21-32051R1

Dear Dr. Boddaert,

We’re pleased to inform you that your manuscript has been judged scientifically suitable for publication and will be formally accepted for publication once it meets all outstanding technical requirements.

Kind regards,

Tim Luckett

Academic Editor

PLOS ONE

Additional Editor Comments (optional):

Reviewers' comments:

Reviewer's Responses to Questions

**Comments to the Author**

1. If the authors have adequately addressed your comments raised in a previous round of review and you feel that this manuscript is now acceptable for publication, you may indicate that here to bypass the “Comments to the Author” section, enter your conflict of interest statement in the “Confidential to Editor” section, and submit your "Accept" recommendation.

Reviewer #1: All comments have been addressed

Reviewer #2: All comments have been addressed

2. Is the manuscript technically sound, and do the data support the conclusions?

Reviewer #1: Yes

Reviewer #2: Yes

3. Has the statistical analysis been performed appropriately and rigorously? 

Reviewer #1: Yes

Reviewer #2: Yes

4. Have the authors made all data underlying the findings in their manuscript fully available?

Reviewer #1: Yes

Reviewer #2: Yes

5. Is the manuscript presented in an intelligible fashion and written in standard English?

Reviewer #1: Yes

Reviewer #2: Yes

6. Review Comments to the Author

Reviewer #1: The authors have adequately addressed all my previous comments. This is a very worthwhile and well presented paper

Reviewer #2: Dear Authors, Thank you for your amendments. I believe the paper has been strengthened and is well presented. I wish you well with the publication of this manuscript.

7. PLOS authors have the option to publish the peer review history of their article (what does this mean?). If published, this will include your full peer review and any attached files.

Reviewer #1: No

Reviewer #2: No

---

## [Editor Report · Acceptance letter]

14 Mar 2022

PONE-D-21-32051R1 

Development of a national quality framework for palliative care in a mixed generalist and specialist care model: A whole-sector approach and a modified Delphi technique 

Dear Dr. Boddaert:

I'm pleased to inform you that your manuscript has been deemed suitable for publication in PLOS ONE. Congratulations! Your manuscript is now with our production department. 

Kind regards, 

on behalf of

Dr. Tim Luckett 

Academic Editor

PLOS ONE